# Evidence-Based Practices and Use among Employees and Students at an Austrian Medical University

**DOI:** 10.3390/jcm10194438

**Published:** 2021-09-27

**Authors:** Daniela Haluza, David Jungwirth, Susanne Gahbauer

**Affiliations:** 1Center for Public Health, Department of Environmental Health, Medical University Vienna, Kinderspitalgasse 15, 1090 Vienna, Austria; davidjungwirth@gmx.at; 2Center for Public Health, Department of Social and Preventive Medicine, Medical University Vienna, Kinderspitalgasse 15, 1090 Vienna, Austria; susanne.gahbauer@meduniwien.ac.at

**Keywords:** medical education, training, hospital staff, clinical practice guidelines, students, capability beliefs

## Abstract

Developed in the pre-internet era in the early 1980s, empirical medical practice, i.e., evidence-based practice (EBP) has become crucial in critical thinking and statistical reasoning at the point-of-care. As little evidence is available so far on how EBP is perceived in the Austrian academic context, we conducted a cross-sectional online survey among a nonrandom purposive sample of employees and students at the Medical University Vienna, Austria (total *n* = 1247, 59.8% females). The German questionnaire assessed both EBP capability beliefs and EBP use, with the respective indices both yielding good internal consistency. We conducted subgroup comparisons between employees (*n* = 638) and students (*n* = 609). In line with Bandura’s self-efficacy theory, we found a correlation between EBP capability beliefs and EBP use, with higher scores reported in the employee group. The results indicated that the participants did not strictly follow the sequential EBP steps as grounded in the item-response theory. Since its emergence, EBP has struggled to overcome the dominating traditional way of conducting medicine, which is also known as eminence-based medicine, where ad hoc decisions are based upon expert opinions, and nowadays frequently supplemented by quick online searches. Medical staff and supervisors of medical students should be aware of the existing overlaps and synergies of these potentially equivalent factors in clinical care. There is a need for intensifying the public and scientific debate on how to deal with the divergence between EBP theory and EBP practice.

## 1. Introduction

About 40 years ago, the realization that results from scientific research did not consistently find their way into everyday clinical care led to the introduction of empirical medical practice, i.e., evidence-based practice (EBP) [1]. In contrast to relying on expert opinions not based on empirical studies (eminence-based medicine), EBP focuses on critical appraisal, systematic literature reviews, and high quality national or even international scientific standards [2,3]. Theoretically, this patient-oriented, contextual process of evidence collection, collaboration, and critical reasoning allows for solid clinical decisions. EBP thus permanently strives toward increasing positive patient outcomes using applied knowledge of “what works best” in the ongoing challenge to improve care, following these steps: start with a defined question, seek out the relevant knowledge, critically evaluate this knowledge, and apply this evaluation’s findings in clinical practice.

Although the importance of EBP is widely accepted, there is a constant debate in the scientific community on how to achieve it. Authors from various medical fields—from statistics to psychosomatics—have suggested different approaches and definitions to address the perceived institutional as well as individual challenges for implementing and conducting EBP [4]. For example, Musalek postulated the necessity to integrate evidence-based medicine in the concept of human-based medicine, allowing for more subjectivity in an attempt to acquire a holistic view on patient needs [5]. Horwitz et al. suggested using the term medicine-based evidence, which represents more than just wordplay, by contradicting EBP in such a way that it is concentrated on clinical practice [6]. Medicine-based evidence means that longitudinal patient profiles, i.e., the biological, clinical, psychological, and social environmental history that match the index patient profile, provide the comparative empirical base for specific case management.

Still, the main quest for EBP is to develop a coherent decision-making theory in view of related science disciplines, given the complexity of physiological and pathological processes and the continuous shift between states of health and disease. The transition from the formerly dominating traditional way to conduct medicine, eminence-based medicine, is an ongoing process that will probably never be completed in view of real-life clinical care and unpredictable emergency situations.

Already in 1999, Isaacs and Fitzgerald suggested seven alternatives to evidence-based medicine in the renowned British Medical Journal, namely, eminence, vehemence, eloquence, providence, diffidence, nervousness, and confidence-based medicine [7]. Although meant as a sarcastic comment, it clearly highlights the gap between theory and practice in everyday medicine. Especially with the need to solve complex surgical and clinical challenges in the specific environment of making decisions under uncertainty in medicine, the notion that medicine is art (the art of healing) rather than science becomes a fact, with the EBP guidelines successfully substituted with senior expert intuition [8,9]. So, it seems that concepts such as EBP, human-based medicine, and eminence-based medicine are not mutually exclusive in medical practice.

As already mentioned, we have seen several proposed rewordings ranging from medicine-based and human-based medicine, and praise for decisions based on extrapolation as well as calls for not abolishing the tradition of eminence-based medicine, which is a well-known concept in the Austrian medical school looking back on a tradition of several hundred years [5,10,11]. In the last decades, the digitalization of information access, mirrored in the constantly rising use of smartphone-based medical applications at point-of-care, has opened up new opportunities to provide (online) evidence-based rather than eminence-based healthcare [8,12,13]. More recently published concepts argue for integrating or even replacing evidence-based medicine with theory-driven and co-designed-based approaches [11]. So, the field is constantly evolving, but also returning to its more traditional roots while failing to connect the loose ends of global disciplines.

Coined by Albert Bandura, the concept of beliefs about capabilities, or self-efficacy, originates from social cognitive psychology and is linked to intentions and behaviors [14]. For measuring individual beliefs about one´s capability to use the components of EBP, the EBP capability beliefs scale (EBPCBS) is widely used. It was originally developed using a Swedish sample of registered nurses, and is considered a valid and reliable tool that can be used on a one-dimensional scale [15]. The tool has been tested in studies among nurses, health professionals, medical students, physical therapists, pharmacists, and social workers in different countries [16,17]). Mirroring the notion that EBP is a step-by-step process, the measure of EBP capability beliefs takes into account the sequential order and increasing complexity of the task. Thus, for the purpose of ability assessment as undertaken using the EBPCBS, the item response theory (e.g., as proposed in the Rasch model), is applicable, which is a widely used psychometric model for analyzing categorical data while balancing respondent skills and item difficulty [18].

Nowadays, the concept of EBP has become crucial for educating medical students and training young clinicians in critical thinking and statistical reasoning when practicing medicine [2,3,8,19]. Considering that EBP is a well-known concept in academic medicine and perceived as the backbone of high-quality clinical care, very little is known of how the current academic discussion and societal trends affect EBP in the Austrian healthcare setting. Using the EBSCBS, this study focused on (i) collecting data on the self-perceived levels of capability beliefs and use, and (ii) comparing these measures between employees and students at the Medical University Vienna, Austria.

## 2. Methods

### 2.1. Study Design

This cross-sectional study surveyed a nonrandom purposive sample of German-speaking employees and students at the Medical University Vienna, Austria. Employees mainly consisted of medical doctors working at the main teaching hospital of the university, i.e., the General Hospital of Vienna, which is part of the university campus of the publicly funded university [20]. The study protocol was approved by the Ethics Committee (18 January 2019) and by the Data Protection Committee of the Medical University Vienna (5 April 2019), and was conducted following the principles of the Declaration of Helsinki and in accordance with the principles of the General Data Protection Regulation.

To ensure general comprehensibility as well as face and content validity of the survey, we performed a paper–pencil-based pre-test among the subjects with (*n* = 13) and without (*n* = 12) a medical background and integrated their feedback in the German questionnaire. The final survey was accessible online from 7 April to 6 May 2019 via Medcampus, the university’s password-protected information management system used for research and teaching administration, which also offered an electronic web-based survey service. The survey was not further advertised, as the Medcampus system directly approached all employees and enrolled students at the university by email. These potential participants had single access to the survey and received an email invitation to participate with a reminder notification two weeks after initial contact. We did not offer incentives for participation. All responses were anonymous and study participation was voluntary. Participants gave their explicit informed consent when sharing their opinions in the survey.

### 2.2. Study Questionnaire

The online survey in German consisted of a general section assessing demographic data on the professional group (employee, student), gender (female, male), age (ten age groups from <20 to 60+ years of age), and seniority level, i.e., year in profession (for employees) or years of studies (for students). In a further section, we assessed the self-reported capability beliefs of EBP using the Evidence-Based Practice Capability Beliefs Scale (EBPCBS) developed and validated by Wallin et al. [15]. After reviewing the available tools for surveying EBP capability beliefs and EBP use, we decided to employ this tool in our study due to its practicability, conciseness, and validity in similar study populations [15,16,17]. The six EBPCBS items reflect the EBP process: (1) formulate questions (about clinical practice to search for new research-based knowledge), (2) search databases, (3) search other (information) sources (e.g., books, journals or ask colleagues), (4) appraise research reports, (5) implement knowledge (and thus contribute to change in clinical practice), and (6) evaluate practice (to assess whether clinical practice is based on research knowledge). The respondents were asked to rate their capability beliefs in every step of the EBP process (EBPCB) on a scale ranging from 1 to 5 (poor to very good). We also asked participants to indicate how often they carried out these EBP activities (EBP use) on a scale ranging from 1 to 5 (rarely/never, about once a year, about once every six months, about once a month, several times a month). Lastly, the open-ended question “Is there anything else you want to tell us?” asked for additional comments from the study population in a text comment box.

### 2.3. Statistical Analyses

We conducted all statistical analyses in SPSS Statistics for Windows (Version 25.0. IBM Corp. Armonk, NY, USA). A two-sided level of significance was set to *p* < 0.05. Missing values were tolerated without interpolation approaches, explaining the deviations from 100% of the total study population. We generated two indices, namely, the EBP capability beliefs index to measure the respondents’ capability beliefs on the EBP process and the EBP use index on the respondents’ use of the EBP process. The six items in the respective scales were summarized and divided by 6 with lower ratings indicating lower levels of agreement. Cronbach’s alpha values of 0.872 (employees: 0.874, students: 0.860) for the EBP capability beliefs index and 0.847 (employees: 0.864, students: 0.828) for the EBP use index indicated a good internal consistency for both indices.

We used descriptive statistics to summarize the quantitative data by reporting percentages, means, and standard deviations (SD). We used T tests to assess the differences between the professional groups. To evaluate the relationships between the three variables, i.e., the two EBP indices and the seniority level, we performed independent correlation tests (Pearson’s correlation coefficient, r) for employees (years in profession) and students (year of studies). As for the free text comments, we conducted a qualitative content analysis of the responses by assigning the narrative answers to different categories and stratified the narrative answers from the two professional groups.

## 3. Results

Although 1253 participants (644 employees and 609 students) responded, we excluded six employees (three males and three females) who did not fill out responses for both EBPCB and EBP use, resulting in a study sample in total of 1247 participants, i.e., 638 employees and 609 students (51.2% vs. 48.8% of the total study sample). Table 1 shows the socio-demographic characteristics of the study sample. As expected, the employees were older than the students, with more females (employees: 63.9%, students: 56.7%) in both academic groups. More than half of the employees reported having more than ten years in the profession, whereas in the student group, first (28.5%) and second (19.4%) year students were most common.

Table 2 depicts the results of the statistical comparison using T tests between employees and students in EBP capability beliefs and use. The employee group rated a mean value of 3.7 on the EBP capability beliefs index, scoring between 3.4 (*Appraise research reports*) and 4.1 (*Search other sources*) on the six EBP activities. The student group rated a mean value of 3.4 on the EBP capability beliefs index, scoring between 3.1 (*Appraise research reports*) and 3.9 (*Search other sources*). The employee group rated a mean value of 3.5 on the EBP use index, scoring between 3.0 (*Appraise research reports*) and 4.4 (*Search other sources*) on the six EBP activities. The student group rated a mean value of 3.2 on the EBP use index, scoring between 2.5 (*Implement knowledge*) and 4.6 (*Search other sources*). The professional groups differed statistically significantly in all items and the two indices (all *p* < 0.05), with the students scoring lower in all items of EBP use except for *Search other sources*.

We examined the correlation between the EBP capability beliefs index, EBP use index, and seniority level (Table 3). We found a strong relationship between the two EBP indices for both professional groups (r > 0.6, *p* < 0.001), whereas the number of years in profession did not correlate with the indices in the employees, but the year of studies (students) showed a negligible, hence statistically significant correlation with the EBP use index (r = 0.1, *p* < 0.001).

In total, 118 participants (9.4%), 59 employees and 59 students, provided additional material in the text comment box. Eight study subjects (one employee and seven students) responding “no” were not considered for further analysis. Qualitative context analysis revealed that six participants found praising words for the survey, whereas the majority of 52 participants (27 employees and 25 students) criticized aspects of the survey content. Specifically, 35 participants (15 employees and 20 students) criticized the EBP items.

Examples of quotes from employees are:

1. “Some items are formulated in a strange way. Use other sources of information (e.g., books, magazines, colleagues)—everyone talks to colleagues about cases, etc. Contribute to changing the professional activity by including the (current) level of knowledge—if my activity is not based on my knowledge, what then? Assess whether the professional activity is based on research findings—if this means questioning current actions and comparing them with current research findings, it is not 100% clear.”

2. “What is meant by ’evaluation of research reports?”

3. “I do not understand the points ’Contributing to the change in professional activity by including the (current) state of knowledge’ and ’Assessing whether the professional activity is based on research findings’. What does this mean?”

Examples for quotes by students are:

1. “I found some questions difficult to understand. For example, I am unsure whether the question ’Assess whether clinical practice is based on research findings’ could be answered by a medical student with anything other than Never, since one does not teach this. The situation is similar with ’Formulating questions about student activity to search for new research-based knowledge’ or ’Contributing to the change in student activity by including the (current) state of knowledge.”

2. “Formulating questions to search for new, research-based knowledge—this wording is very difficult for me to understand. I find the question interesting, because I often have the feeling that studying at the Medical University Vienna is 100% science and 0% nature. I appreciate the scientific work, but would also like to see nature included, for example an elective course on Traditional European Medicine.”

## 4. Discussion

EBP has contributed substantially to the improvement of the quality of research by transparently documenting the problems with existing research and subsequently developing better research standards [9,21]. EBP has also improved the practice of medicine by developing methods and techniques for generating systematic literature reviews and clinical practice guidelines. To improve clinical outcomes and patient safety, and to reduce healthcare costs, implementing EBP is critical. As far as we know, this study is the first one collecting data on respective practices and beliefs among healthcare professionals and medical students at an Austrian medical university. The present study revealed differences in capability beliefs to perform EBP among employees and students in an academic setting. The original term EBP is constantly evolving and being reworded, ranging from medicine-based and human-based medicine to suit the specific requirements of different scientific disciplines. In our study, we did not strive to create another idea or concept of EBP, as we used an already well-known and validated survey tool.

A Chinese study found that about three quarters of physicians regularly applied EBM in their routine daily practice [22]. However, a review showed that physicians (both in in- and out-patient settings) encountered various obstacles when using EBP in clinical practice, including a lack of time, EBP skills, patient-related factors, and their own attitudes [4]. Murphy et al. found a lack of practical guidance tailored to students and their supervisors during clinical placements, potentially decreasing their capability and willingness for using EBP [21]. Organizational culture for EBP was significantly and positively related to EBP beliefs and EBP implementation among health professionals in the U.S.A. [19]. In our study, the two professional groups differed in their reported EBP capability beliefs and use, with higher ratings observed among employees. This result is in contrast to a Swedish study using the same assessment tool that reported opposite results with higher ratings (except for the item *Search other sources*) among students compared to health professionals [23]. However, the differences in EBP capability beliefs in this study were not statistically significant and the study population was quite small. In our study, we found higher ratings for *Search other sources* among students for EBP use. Contradicting findings in different study populations and different countries could highlight the role of organization-specific educators in promoting an EBP culture, e.g., by adapting the medical curricula and postgraduate programs. So, establishing an EBP culture among educators might affect the EBP capability beliefs and future intentions of students to adopt EBP in clinical routines during their clinical placement [23].

In line with other authors, we found that high EBP capability beliefs were strongly correlated with more frequent EBP use [16,19]. The number of years in profession did not correlate with both EBP indices in the employee group. Among students, the year of studies only showed a negligible correlation with the EBP use index, and this statistical significance could be interpreted as being sample size related. In accordance with other studies, the students reported medium scale EBP capability beliefs and lower scale EBP use on the items *Implement knowledge* and *Evaluate practice* [17,23]. As these are concluding steps of the EBP process, it seems that medical students would profit from educational and organizational support in these activities.

In the present study, we found moderate and rather high capability beliefs to perform and use EBP among both employees and students, with the items *Search databases* and *Search other sources* as dominant steps in the EBP process according to the participants. This result is in agreement with a review on teaching EBP, suggesting a high EBP competence among health students and also an increasing use of EBP promoting technology, i.e., mobile devices, simulations, and web content [24]. Our observations could be interpreted as a general trend towards increased online health information seeking behavior and the use of electronic devices among all strata of the Austrian population, as observed worldwide [12,13]. Notably, the constant availability of digital content could also impact capability beliefs and use of EBP among medical students representing the so-called digital natives, i.e., those individuals born after 1980 [25,26]. This aspect seems to be neglected in the tools for assessing progress and efficiency regarding the research, teaching, and evaluation of EBP practices in a stepwise manner. The EBPCBS instrument used in our study, as well as other tools and conceptual frameworks, was grounded on the concept of an inherent understanding of just these subsequent steps [3,15,27]. However, in theory the step-by-step process would argue for a more evenly distributed ratings picture, without overlooking the first EBP step *Formulate questions*, which is potentially perceived as being more banal and superfluous in everyday clinical practice, especially amongst more experienced health professionals [23,28]. Moreover, this might be the turning point where concepts of eminence-based medicine, intuition, and seniority take over, simply leading to a strict following of the process to absurdity [8,9].

In the pertinent international literature, researchers use heterogeneous EBP assessment tools, constantly design new ones in different languages, and test them in various study populations and clinical settings [19,27]. This limits the comparability of the retrieved results and usefulness of the tools, hampering the advancement of EBP for clinical practice and (continuous) medical education. Although we used the validated EBPCBS in our survey, qualitative context analysis of free text comments indicated the possibility of potential sources of misunderstanding in the wording of the scale. In addition, the stepwise EBP process, or as such following the item response theory, was violated when taking into account the answers of the participants [18]. This was unexpected and should motivate future in-depth qualitative studies assessing for contextual factors influencing the capability beliefs and use of EBP.

Time resources for educators for continuous teaching and training as well as for regular evaluations and feedback loops are key for supervising students. Likewise, students also need time to practice recently acquired medical skills and knowledge. So far, little is known on the interconnection between EBP and the hardly measurable concepts referring to eminence-based medicine or senior expert intuition [7]. A sound EBP culture could eventually assist in raising awareness for the existence of these concepts, helping former students who enter the professional world to understand the influence of hierarchy in everyday clinical decision making [8,9]. To establish an organizational EBP culture ensuring an excellent quality of care for patients, our findings suggest that medical curricula should be screened and adapted, and supported by high-level clinical management initiatives and sustainable strategies.

As for the study limitations, the cross-sectional nature of our study did not allow for any causal attributions. Participation rates were high in view of previous Medcampus surveys [25,29] and the total number of participants of 1247 was significantly higher [22,23] or similar [16,30,31] to related studies. Employees of the Medical University Vienna that were the recipients of the invitation to participate in this survey mainly consisted of medical doctors and only very few representatives of other health professions, such as biologists, physiotherapists or psychologists. Similarly, the vast majority of enrolled students were medical students, with very few exceptions such as medical informatics students. Following the directives of the local Data Protection Committee, we were not able to ask for specific sub-professions in the questionnaire to protect the identity of the underrepresented professions and studies. To address this issue, we concretely asked for opinions of medical doctors and medical students in the invitation letter for study participation.

Only German-speaking, highly educated people affiliated with the Medical University Vienna were eligible to participate in this study. This limits the generalizability of the study results to other sections of academic and especially non-academic populations. Nevertheless, this study aimed at assessing EBP practices in an academic German-speaking study population affiliated with the largest medical university in Austria. Given the mono-centric design of this study, existing sub-cultures and traditions may limit the direct implications for other universities and clinics. Future studies should assess contextual factors to explore the potential organizational differences for the teaching and use of EBP between different health professional groups, also testing for potential differences with respect to gender and diversity.

The study questionnaire assessed the self-reported capability beliefs of EBP using a validated scale, and a pre-test was also conducted [15]. However, the qualitative context analysis of free text comments revealed potential sources of misunderstanding in the wording of the items, as expressed by quite a small number of study subjects. The reasons for this should be further evaluated by face-to-face interviews among employees and students to differentiate between an unfamiliarity of the terminology and the EBP process itself. Furthermore, to advance this field of research in a changing and demanding profession, our findings suggest elucidating whether a survey is the appropriate format to collect data on EBP capability belief and use, and whether the scale would be a valid and reliable measure despite good psychometric properties [32]. Additionally, it was beyond the scope of this study to assess the impact of non-evidence-based practices and beliefs, such as alternative medicine in general or Traditional European Medicine specifically, as stated in one of the free text comments. Methods including Delphi surveys, mixed-method approaches, workshops, and longitudinal assessments might be useful to complement cross-sectional surveys in this context.

## 5. Conclusions

The present study compared the self-reported EBP capability beliefs and use between health professionals and students in an academic setting in Austria. In line with Bandura’s self-efficacy theory, we found a correlation between EBP capability beliefs and use. Potentially picturing the ongoing digitalization in the field of medicine, the results indicate that the study participants do not strictly follow the sequential order of the EBP steps as grounded in the item response theory. Thus, our results suggest a profound re-thinking of the established methodological ways of surveying for EBP among current and future healthcare professionals while addressing the complexity of EBP in a strenuous clinical setting.

The identified difference in the capability beliefs and use of EBP among employees and students highlights the need for intensifying—or even beginning—the public and scientific debate on how to balance the divergence between EBP theory, i.e., research, and EBP practice, i.e., implementation. Medical staff and the educators of medical students should be aware of the existing overlaps and synergies of other important, potentially equivalent factors in clinical care, particularly in eminence-based medicine, and the immense impact of a nearly unlimited access to online content in modern clinical decision making.

## Figures and Tables

**Table 1 jcm-10-04438-t001:** Socio-demographic characteristics of study participants (*n* = 1247), stratified by professional group (employees vs. students).

	Employees (*n* = 638)		Students (*n* = 609)	
	*n*	%	*n*	%
Age				
<20 years	2	0.3	27	4.4
20–24 years	14	2.2	336	55.2
25–29 years	74	11.6	167	27.4
30–34 years	86	13.5	49	8.0
35–39 years	103	16.2	30	4.9
40–44 years	64	10.0		
45–49 years	100	15.7		
50–54 years	77	12.1		
55–59 years	72	11.3		
60+ years	45	7.1		
Total	637	100.0	609	100
Gender				
Female	404	63.9	345	56.7
Male	228	36.1	264	43.3
Total	632	100.0	609	100.0
Years in profession (employees)				
<5 years	201	31.7		
5–10 years	108	17.0		
>10 years	325	51.3		
Total	634	100.0		
Year of studies (students)				
1st year			172	28.5
2nd year			117	19.4
3rd year			73	12.1
4th year			81	13.4
5th year			69	11.4
6th year and more			92	15.2
Total			604	100

**Table 2 jcm-10-04438-t002:** Comparison of reported evidence-based practice (EBP) capability beliefs and use between employees and students (total *n* = 1247).

	Employees (*n* = 638)		Students (*n* = 609)		
	Mean	SD	Mean	SD	*p*-Value ^1^
EBP capability beliefs					
EBP capability beliefs index	3.70	0.86	3.35	0.80	<0.0001
Formulate questions	3.53	1.10	3.18	1.05	<0.0001
Search databases	3.99	1.05	3.65	1.05	<0.0001
Search other sources	4.09	0.93	3.92	0.96	0.002
Appraise research reports	3.44	1.17	3.05	1.01	<0.0001
Implement knowledge	3.61	1.11	3.11	1.06	<0.0001
Evaluate practice	3.55	1.19	3.16	1.09	<0.0001
EBP use					
EBP use index	3.53	1.13	3.17	1.03	<0.0001
Formulate questions	3.09	1.62	2.61	1.50	<0.0001
Search databases	4.26	1.27	4.08	1.25	0.016
Search other sources	4.39	1.12	4.59	1.01	0.001
Appraise research reports	2.98	1.61	2.71	1.49	0.002
Implement knowledge	3.37	1.49	2.45	1.54	<0.0001
Evaluate practice	3.07	1.62	2.58	1.54	<0.0001

^1^ All subgroup differences (employees vs. students) statistically significant (*p* < 0.05) as calculated by T tests.

**Table 3 jcm-10-04438-t003:** Correlations between the evidence-based practice (EBP) capability beliefs index, evidence-based practice use index and years in profession (employees) and year of studies (students).

	Years in Profession (Employees)	EBP Capability Beliefs	EBP Use Index
Years in profession (employees)	1		
EBP capability beliefs	r = 0.009, *p* = 0.822	1	
EBP use index	r = 0.015, *p* = 0.699	r = 0.617, *p* < 0.0001 ^1^	1
	**Year of Studies (Students)**	**EBP Capability Beliefs**	**EBP Use Index**
Year of studies (students)	1		
EBP capability beliefs	r = 0.059, *p* = 0.149	1	
EBP use index	r = 0.131, *p* = 0.001 ^1^	r = 0.626, *p* < 0.0001 ^1^	1

^1^ Correlation significant (*p* < 0.05), r: Pearson’s correlation coefficient.

## Data Availability

The datasets analyzed during the current study are available from the corresponding author on reasonable request.

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
