# Peer review of "Evidence-Based Practices and Use among Employees and Students at an Austrian Medical University"

_jcm, 2021, doi:10.3390/jcm10194438_

Round 1

Reviewer 1 Report

General comment:

My responsibility as a reviewer is, on the one hand, to check the present manuscript for rigor, but also to elaborate on very basic questions of scientific conduct. This is a very time-consuming undertaking, which I am willing to carry out, but I also expect this commitment from the authors of the respective manuscript. The deletion of criticized paragraphs with the reference to avoid confusion of the reader is by no means sufficient. Rather, I expect a content-related discussion, since the authors will certainly have thought of something when creating these paragraphs. If these paragraphs are now deleted without replacement, the question arises as to why they were not omitted in the first place and whether the synthesis of the data was done with the appropriate care. The same applies to the deletion of variables. If variables are deleted to avoid the discussion of content, I wonder why this variable was surveyed at all? It seems to me that there was no prior hypothesis generation regarding these variables, which I don't think is good scientific practice.

Major comments:

We agree and modified the specific phrase as follows:
Authors from various medical fields - from statistics to psychosomatics – suggested different approaches and definitions to address perceived institutional as well as individual challenges for implementing and conducting EBM.

This is not a specification rather the opposite – the authors did not elaborate on the raised concern, rather changed the paragraph content-wise.

We agree and modified the specific phrase as follows and hope that it is now clearer, that we see the digitalization of nowadays clinical medicine as a further step, but also a counterpart of the traditional way to retrieve information in medicine.

However, in the last decades, the digitalization of information access has opened up new opportunities to provide objective evidence-based rather than eminence-based healthcare.

Why is this the case? An example would help here.

We are very thankful for this comment. We agree that it might be that this aspect is only interesting from a national point of view. In response to this request, we deleted the according paragraph. The justification of using the EBMCBS is already provided in the preceding paragraph.

The response is lacking the justification for using this one specific tool. As there seem many other tools out there, I think it would be fair to explain why the authors did what they did.

Thank you for raising this issue. Due to ethical reasons including the Data Protection Committee, we were not able to include the granular item on specific sub-professions in the questionnaire. This is because there are only very few non-physicians among the staff, i.e. mainly medical doctors among a negligible proportion of other professions. In the invitation letter for study participation, we concretely asked for opinions of medical doctors.

I think there is a misunderstanding between ‘staff’ and ‘recipients of the survey vial the mailing list’. In my own experience physicians are not the most common staff in a university hospital. However, it could be the case that the mailing list is only accessible for physicians. If so, I would ask the authors to state that in the manuscript. If not, I would like to ask the authors to clarify on that.

As you are stating in the discussion section later there are mainly medical students included, but also some students enrolled in medical informatics. How were you able to know that if the data protection committee did not allow to collect this data? And did you include these students in the analysis? If so, why? It would have been easy to exclude them as they increase bias in the sample.

In response to this request, we deleted the information on residence, to avoid confusion in readers.

If you can delete this variable now, why did you include it in the first place? What was the hypothesis for these variables?

To ensure general comprehensibility as well as face and content validity of the survey, we performed a paper-pencil-based pre-test among subjects with (n = 13) and without medical background (n = 12), and integrated their feedback in the German questionnaire.

I appreciate this addition.

We agree on the necessity to include gender and diversity aspects in our survey. As requested by the ethics committee, we indeed asked for diverse gender, too, but we used the binary sex/gender variable automatically derived from the Medcampus system for accuracy and statistical reasons (i.e. binary variable).

Why is a binary variable more accurate? What is the influence of diverse gender on statistics? This is not clear to me. Indeed, it would have been totally fine for me to hear that you did not consider this at first place and will do this in case of a follow up survey.

Thank you for raising this issue. We respond to it by changing the paragraph accordingly and now focus on the more relevant professional group differences, rather than gender differences we found.

Why are gender differences not relevant? You have a ‘gender medicine unit’ on campus – obviously it is relevant. Deleting a paragraph to avoid content-related discussion is not sufficient.

These opposite findings highlight the essential role of organization-specific educators in promoting EBP culture, e.g. by adapting the medical curricular. So, establishing an EBP culture among educators might effect EBP capability beliefs and future intentions to adopt EBP in clinical routine during clinical placement among students.

So, you say it is context-specific? Swedish EBP is different than Austrian EBP? Why?

Reviewer comment:
Also, it would be interesting to elaborate on the EBM use score for ‘Search other sources’ as medical students score higher than their healthcare professional counterparts.

This statement refers to the Swedish study, it was not found in our study.

I refer to table 2 ‘EBP use’. In the revised document you state that employees scored 4.39 while students scored 4.59. Therefore, it refers to your study and not the Swedish one. Please elaborate on that.

To avoid confusing, we deleted the phrase in Traditional European Medicine.

In fact, the deletion of this sentence is alarming to me. It leaves the student's statement incomplete and at the same time shows that authors shy away from a substantive discussion. This statement of the student says something important about the sample and how valid the collected data is. Bias is the biggest problem in online surveys. That is why it is so important to notice it and discuss it adequately. Deleting this sentence is therefore not only simply avoiding discussion, but also withholding data that negatively affects the overall message of the study.

Besides that, the correct citation would have required ‘[…]’ to let the reader know that the authors shortened the answer.

Minor comment:

There is a number error in the first line of the abstract

As the authors made the full reference list available only now, I would like to add the following: Eleven references are self-citations and most of these are only used once in a bulk citation (Line 352). As the discussed observation is only a minor finding and the authors only speculate about what this result could mean, but without elucidating for and against, I wonder if such a number of citations is justified.

Reviewer 2 Report

I would thanks the editor for the reviewing proposed. This manuscript present an importante topic about EBV. In general, this article is quite difficulte to undertand and should be both reduced and reformulated to help his comprehension. The comparison between employe and student is questionable. The research question that they want to explore is not clearly understandable. Stastistical analysis seems suitable. The tables are difficult to read and should be reformated/simplified. In conclusion, this article deal with an important topic but the scientific soundnes remain poor. Best regards.

Author Response

I would thanks the editor for the reviewing proposed. This manuscript present an importante topic about EBV. In general, this article is quite difficulte to undertand and should be both reduced and reformulated to help his comprehension. The comparison between employe and student is questionable. The research question that they want to explore is not clearly understandable. Stastistical analysis seems suitable. The tables are difficult to read and should be reformated/simplified. In conclusion, this article deal with an important topic but the scientific soundnes remain poor. Best regards.

Response:

We appreciate the in general favorable evaluation of our article. Please be aware that the topic is evidence-based practice (EBP). Honestly, we miss a little more detailed and especially constructive guideline for how we could further increase comprehensibility of our research work.

Please note that the comparison between employees/health professionals etc. and students is a well-known, relevant procedure in many scientific disciplines, not only in EBP studies, ranging from medical education to organizational and management studies. Here are only a few examples of countless papers:

Boström, A.-M.; Sommerfeld, D.K.; Stenhols, A.W.; Kiessling, A. Capability beliefs on, and use of evidence-based practice among four health professional and student groups in geriatric care: A cross sectional study. PloS one 2018, 13, e0192017.

Kyriakoulis, K.; Patelarou, A.; Laliotis, A.; Wan, A.C.; Matalliotakis, M.; Tsiou, C.; Patelarou, E. Educational strategies for teaching evidence-based practice to undergraduate health students: systematic review. J Educ Eval Health Prof 2016, 13, 34-34, doi:10.3352/jeehp.2016.13.34.

Wernhart, A.; Gahbauer, S.; Haluza, D. eHealth and telemedicine: Practices and beliefs among healthcare professionals and medical students at a medical university. PLoS One 2019, 14.

Poorchangizi B, Borhani F, Abbaszadeh A, Mirzaee M, Farokhzadian J. Professional Values of Nurses and Nursing Students: a comparative study. BMC Med Educ. 2019 Nov 27;19(1):438.

Also, please be aware of the high number of participants, i.e. more than 1200, which is rare in surveys in general. We also used a well-known validated research tool grounded on established theories such as the concept of self-efficacy and a protocol that follows relevant ethical and methodological guidelines such as the Cherries checklist for online surveys, and the text is written in accordance with the journal`s guidelines for authors. We also pretested our survey to ensure compensability. In addition, we explained potential limitations in great detail in the respective section.

In response to this request, we modified our tables to enhance readability. We hope that the tables, which will be streamlined and adjusted during the layout for publication anyway, are now easier to understand.

So, we hope that pointing out that our study that is the first of its kind in our study population, has high power, a sound method, and a clear purpose helps to emphasize on the value of our contribution for the scientific community.

This manuscript is a resubmission of an earlier submission. The following is a list of the peer review reports and author responses from that submission.

Round 1

Reviewer 1 Report

Haluza et al. report about a survey performed with healthcare-professionals and medical students about their beliefs and practices about evidence-based medicine (EBM).

Major comments:

Introduction

  • The introduction is meticulously written, giving a very broad overview on EBM. However, due to this width the introduction lacks focus. The authors shed light on various aspects, but do not go into any of them in greater depth., e. g.:
    • Line 45: Authors from various medical fields from statistics to psychosomatics struggle with institutional and individual challenges for implementing and conducting EBM” – Why do they struggle? This is an important information as it could give a rationale for conducting this study. Instead answering this important question, the authors elaborate on the necessity of EBM integration in day-to-day practice which is certainly correct.
    • Line 66: “In the last decades, health technology has opened up new opportunities to provide objective evidence-based rather than eminence-based healthcare” – How can health technology help? Which technology do you refer to? Is this really important for the conduction of the survey?
    • Line 95: The necessity of this paragraph is not clear to me.
      • Why is academization of physiotherapy relevant in this context? Shouldn’t healthcare professionals follow EBM no matter if they have an advanced degree or not?
      • Why is research from other countries not applicable to the Austrian context?
      • The rationale of the study and the methods applied would be much stronger if the authors elaborate on why the picked this particular questionnaire, rather than describing all the potential tools available.

Methods

  • The design and conduction of online surveys is challenging, and I appreciate the efforts the authors already undertook. However, there are some relevant information missing or unclear in the current form of the manuscript:
    • Which profession did the health professionals have? It is imaginable that EBM perception and use is different, e. g. between physiotherapists and physicians. As this could lead to bias, a more granular query of the profession would be desirable. The authors’ statement that ‘mainly medical professionals’ took part in the survey leads to the questions: Is mainly 70% or 90%? What are the other professions? How could you discriminate? Did you include a more granular query? If so, the methods/results section would benefit from this addition.
    • The same aspect is valid for ‘students’. Implicitly we are talking about medical students. But as there are surely students from other biomedical areas on campus it would be good to know explicitly.
    • Why is the residence of participants important? Is there a rationale for asking this particular socio-demographic variable? How are ‘urban’ and ‘rural’ defined?
    • Why is the size of the hospital relevant? in fact, shown like this, it diminishes the significance of the study as ‘only’ 6% of the student population are included. I understand that this population is hard to recruit and therefore I like to state that the sample-size is encouraging and is certainly large enough to draw some conclusions.
    • It would be interesting to know how big the pilot-size was, if they had the possibility to give feedback and if the feedback lead to changes in the survey.
    • In times of the acknowledgment of diversity I recommend an additional gender option beside ‘male’ and ‘female’

Results

  • Is the study population representative for the whole population? At least in terms of gender and age this should be easily done. 56.7% female students seem relatively low as the average female students in German speaking countries range between 60-70%.
  • The authors state, that ‘…year of studies (students) showed a small, but statistically significant correlation with the EBP use index (r=0.1, p<0.001)’. Even though I agree that this is statistically significant I do not think it is correlating. The major contributor to the strong p-value is the sample-size rather than a correlation.

Discussion

  • Interestingly, the authors found a difference between female and male participants on employee and whole-sample level. However, analysis of this difference is implicit and focus on the non-existent difference in the student population, rather than discussing the finding per se. Further, connection between this finding and the ‘gender-medicine-unit’ is unclear. Although it is commendable that such a unit has been established its influence on EBM use and practice is obscure.
  • Line 300: This citation should be mentioned in the introduction as it explains the struggles of health professionals with EBM.
  • How do you explain the contradictory results of the Swedish study?
  • Also, it would be interesting to elaborate on the EBM use score for ‘Search other sources’ as medical students score higher than their healthcare professional counterparts.
  • I think it would be great to get more information about the consequences of this study to further research and education. It is not sufficient to write students need more education on EBM. There should be concrete ideas how to achieve a better understanding and application of EBM. On the other hand, one of the two cited students already stated that it is questionable if this survey is applicable to students, as they do not have to perform EBM in day-to-day clinics. For getting an idea about implementation of EBM in clinics it would be much more informative to ask residents during the residency/fellowship and see if EBM measures develop over time. Of course, the scaling has to be more granular.
  • Even so some limitations are discussed, the authors missed the chance to emphasize bias as a main problem in online surveys. It is important to discuss if the sample included in the study mirrors the sampled population. As it is not randomized we can not be sure about that. Further, the authors could at least try to validate the sample with the existing data. It would be easy to assess proportions of gender and age and compare them with the whole population at the university hospital. Interestingly, the second student cited by the authors suggests bias in terms of representativeness of the sample. Even though the student described problems regarding the wording, the focus of this comment is not the study by itself but on the lack of medical curricula in Traditional European Medicine. There is for sure a role of EBM perception and use in Traditional European Medicine, but this is not relevant for this survey. Again – the nature of online survey brings uncertainty regarding the participating sample. But this has to be acknowledged and thoroughly discussed.

Minor comments:

  • There are several typos in the manuscript
  • Roughly half of the citations are missing in the bibliography (22-43)
  • Line 220: Students score lower in ‘Implement knowledge’ contrary to what is reported by the authors.
  • Line 299: I would suggest to re-write these brackets, e. g. (in in- and out-patient settings) as GPs are also clinicians.

In conclusion, I appreciate the authors efforts to elucidate health professionals’ and students’ perception and use of EBM. However, for the latter population only very limited rationale is presented which is also mentioned by one of the participants. Why should students use EBM? For sure this is important in their clinical development later on, but at the present stage I am not able to see how they could apply that concept properly.

Further, some important socio-demographic data is missing (which health professionals? Which students?) while the relevance and definition of other variables is not clear and not followed up later on in the manuscript (‘Residence’).

The discussion remains very superficial and did not point out any consequences resulting from the present study. In terms of medical education, the authors suggest educating more on EBM but do not give any ideas how this could be achieved. Consequences for health professionals are not even discussed.

Author Response

Haluza et al. report about a survey performed with healthcare-professionals and medical students about their beliefs and practices about evidence-based medicine (EBM).

Response:

We thank Reviewer 1 for the valuable comments that were very helpful to improve readability and the quality of our article.

Major comments:

Introduction

  • The introduction is meticulously written, giving a very broad overview on EBM. However, due to this width the introduction lacks focus. The authors shed light on various aspects, but do not go into any of them in greater depth., e. g.:
    • Line 45: “Authors from various medical fields from statistics to psychosomatics struggle with institutional and individual challenges for implementing and conducting EBM” – Why do they struggle? This is an important information as it could give a rationale for conducting this study. Instead answering this important question, the authors elaborate on the necessity of EBM integration in day-to-day practice which is certainly correct.

Response:

We agree and modified the specific phrase as follows:

Authors from various medical fields - from statistics to psychosomatics – suggested different approaches and definitions to address perceived institutional as well as individual challenges for implementing and conducting EBP.

    • Line 66: “In the last decades, health technology has opened up new opportunities to provide objective evidence-based rather than eminence-based healthcare” – How can health technology help? Which technology do you refer to? Is this really important for the conduction of the survey?

Response:

We agree and modified the specific phrase as follows and hope that it is now clearer, that we see the digitalization of nowadays clinical medicine as a further step, but also a counterpart of the traditional way to retrieve information in medicine.

However, in the last decades, the digitalization of information access has opened up new opportunities to provide objective evidence-based rather than eminence-based healthcare.

    • Line 95: The necessity of this paragraph is not clear to me.
      • Why is academization of physiotherapy relevant in this context? Shouldn’t healthcare professionals follow EBM no matter if they have an advanced degree or not?
      • Why is research from other countries not applicable to the Austrian context?
      • The rationale of the study and the methods applied would be much stronger if the authors elaborate on why the picked this particular questionnaire, rather than describing all the potential tools available.

Response:

We are very thankful for this comment. We agree that it might be that this aspect is only interesting from a national point of view. In response to this request, we deleted the according paragraph. The justification of using the EBMCBS is already provided in the preceding paragraph.

Methods

  • The design and conduction of online surveys is challenging, and I appreciate the efforts the authors already undertook. However, there are some relevant information missing or unclear in the current form of the manuscript:
    • Which profession did the health professionals have? It is imaginable that EBM perception and use is different, e. g. between physiotherapists and physicians. As this could lead to bias, a more granular query of the profession would be desirable. The authors’ statement that ‘mainly medical professionals’ took part in the survey leads to the questions: Is mainly 70% or 90%? What are the other professions? How could you discriminate? Did you include a more granular query? If so, the methods/results section would benefit from this addition.
    • The same aspect is valid for ‘students’. Implicitly we are talking about medical students. But as there are surely students from other biomedical areas on campus it would be good to know explicitly.

Response:

Thank you for raising this issue. Due to ethical reasons including the Data Protection Committee, we were not able to include the granular item on specific sub-professions in the questionnaire. This is because there are only very few non-physicians among the staff, i.e. mainly medical doctors among a negligible proportion of other professions. In the invitation letter for study participation, we concretely asked for opinions of medical doctors. 

To address this important aspect, we now elaborate on this aspect in the limitations section:

Staff of the Medical University Vienna mainly consisted of medical doctors and only very few representatives of other health professions such as biologists, physiotherapists or psychologists. Similarly, the vast majority of enrolled students are medical students, with very few exceptions such as medical informatics. Following the directives of the local Data Protection Committee, we were not able to ask for specific sub-professions in the questionnaire to protect the identity of the underrepresented professions. In the invitation letter for study participation, we concretely asked for opinions of medical doctors and medical students.

Why is the residence of participants important? Is there a rationale for asking this particular socio-demographic variable? How are ‘urban’ and ‘rural’ defined?

Response:

In response to this request, we deleted the information on residence, to avoid confusion in readers.

    • Why is the size of the hospital relevant? in fact, shown like this, it diminishes the significance of the study as ‘only’ 6% of the student population are included. I understand that this population is hard to recruit and therefore I like to state that the sample-size is encouraging and is certainly large enough to draw some conclusions.

Response:

We agree and deleted the size of the hospital and other related figures.

    • It would be interesting to know how big the pilot-size was, if they had the possibility to give feedback and if the feedback lead to changes in the survey.

Response:

We totally agree and added the following information:

To ensure general comprehensibility as well as face and content validity of the survey, we performed a paper-pencil-based pre-test among subjects with (n = 13) and without medical background (n = 12), and integrated their feedback in the German questionnaire.

    • In times of the acknowledgment of diversity I recommend an additional gender option beside ‘male’ and ‘female’

Response:

We agree on the necessity to include gender and diversity aspects in our survey. As requested by the ethics committee, we indeed asked for diverse gender, too, but we used the binary sex/gender variable automatically derived from the Medcampus system for accuracy and statistical reasons (i.e. binary variable).

Results

  • Is the study population representative for the whole population? At least in terms of gender and age this should be easily done. 56.7% female students seem relatively low as the average female students in German speaking countries range between 60-70%.

Response:

Indeed, we have slightly more male medical students than females in Vienna, which is suggested to be caused by the entrance exam potentially favoring male skills. As we did not focus on gender differences but on the professional groups, we did not discuss the impact on gender further.

  • The authors state, that ‘…year of studies (students) showed a small, but statistically significant correlation with the EBP use index (r=0.1, p<0.001)’. Even though I agree that this is statistically significant I do not think it is correlating. The major contributor to the strong p-value is the sample-size rather than a correlation.

 Response:

We agree and slightly changed the wording:

Results: …year of studies (students) showed a negligible, hence statistical significant correlation with the EBP use index (r=0.1, p<0.001).

Discussion: Among students, year of studies only showed a negligible correlation with the EBP use index, and the statistical significance could be interpreted as sample size-related.

Discussion

  • Interestingly, the authors found a difference between female and male participants on employee and whole-sample level. However, analysis of this difference is implicit and focus on the non-existent difference in the student population, rather than discussing the finding per se. Further, connection between this finding and the ‘gender-medicine-unit’ is unclear. Although it is commendable that such a unit has been established its influence on EBM use and practice is obscure.

Response:

Thank you for raising this issue. We respond to it by changing the paragraph accordingly and now focus on the more relevant professional group differences, rather than gender differences we found.

  • Line 300: This citation should be mentioned in the introduction as it explains the struggles of health professionals with EBM.

Response:

Thank you for this suggestion. We now cite this reference in the intro section.

Authors from various medical fields - from statistics to psychosomatics – suggested different approaches and definitions to address perceived institutional as well as individual challenges for implementing and conducting EBP (ref.).

  • How do you explain the contradictory results of the Swedish study?

Response:

Thank you for raising this issue. We respond to it by changing the paragraph accordingly.

These opposite findings highlight the essential role of organization-specific educators in promoting EBP culture, e.g. by adapting the medical curricular. So, establishing an EBP culture among educators might effect EBP capability beliefs and future intentions to adopt EBP in clinical routine during clinical placement among students.

  • Also, it would be interesting to elaborate on the EBM use score for ‘Search other sources’ as medical students score higher than their healthcare professional counterparts.

Response:

This statement refers to the Swedish study, it was not found in our study. To make this clearer, we slightly modified the phrase:

This result in is contrast with a Swedish study using the same assessment tool and reporting the opposite result with higher ratings (except the item Search other sources) among students compared to health professionals.

  • I think it would be great to get more information about the consequences of this study to further research and education. It is not sufficient to write students need more education on EBM. There should be concrete ideas how to achieve a better understanding and application of EBM. On the other hand, one of the two cited students already stated that it is questionable if this survey is applicable to students, as they do not have to perform EBM in day-to-day clinics. For getting an idea about implementation of EBM in clinics it would be much more informative to ask residents during the residency/fellowship and see if EBM measures develop over time. Of course, the scaling has to be more granular.
  • Even so some limitations are discussed, the authors missed the chance to emphasize bias as a main problem in online surveys. It is important to discuss if the sample included in the study mirrors the sampled population. As it is not randomized we can not be sure about that. Further, the authors could at least try to validate the sample with the existing data. It would be easy to assess proportions of gender and age and compare them with the whole population at the university hospital. Interestingly, the second student cited by the authors suggests bias in terms of representativeness of the sample. Even though the student described problems regarding the wording, the focus of this comment is not the study by itself but on the lack of medical curricula in Traditional European Medicine. There is for sure a role of EBM perception and use in Traditional European Medicine, but this is not relevant for this survey. Again – the nature of online survey brings uncertainty regarding the participating sample. But this has to be acknowledged and thoroughly discussed.

Response:

We totally agree on these two points raised and integrated the implementations of the study findings in the text. As for,

Discussion:

In the pertinent international literature, researchers use heterogeneous EBP assessment tools, constantly design new ones in different languages, and test them in various study populations and clinical settings. This limits comparability of the retrieved results and usefulness of the tools, hampering the advancement of EBP for clinical practice and (continuous) medical education. Although we used the established EBPCBS in our survey, qualitative context analysis of free text comments indicated the possibility of potential sources of misunderstanding in the wording of the scale. In addition, the stepwise EBP process, as such following the item-response theory, was violated when taking into account the answers of the participants [15]. This was unexpected and should motivate for future in-depth qualitative studies assessing contextual factors to teach and use EBP.

Time resources for educators for continuous teaching and training as well as regular evaluations and feedback loops are key for supervising students. Likewise, students also need time to practice implementation and evaluation of new medical skills and knowledge.

So far, little is known on the interconnection between EBP and the hardly measurable concepts referring to eminence-based medicine, or senior expert intuition [8]. A sound EBP culture could eventually assist in raising awareness for the existence of these concepts, helping former students who enter the professional world to under-stand the influence of hierarchy in every-day clinical decision making [9, 10]. To establish an organizational EBP culture ensuring excellent quality care for patients, our findings suggest that medical curricula should be screened and adapted, supported by clinical management initiatives and strategies.

Conclusion:

The present study compared self-reported EBP capability beliefs and use between health professionals and students in an academic setting in Austria. In line with Ban-dura’s self-efficacy theory, we found a correlation between EBP capability beliefs and use. Potentially picturing the ongoing digitalization in medicine, study participants indicated to not strictly follow the sequential order of the EBP steps as grounded in the item-response theory. Thus, our results suggest a profound re-thinking of established methodological ways of surveying EBP among current and future healthcare professionals while addressing the complexity of EBP in a strenuous clinical setting.

Identified difference in capability beliefs and use of EBP among employees and students highlight the need for intensifying – or even start – the public and scientific debate on how to balance the divergence between EBP theory, i.e. research, and EBP practice, i.e. implementation. Medical staff and educators of medical students should be aware of existing overlaps and synergies of other important, potentially equivalent factors in clinical care, particularly eminence-based medicine, and the immense impact of nearly unlimited access to online content in modern clinical decision making.

To avoid confusing, we deleted the phrase in Traditional European Medicine.

Minor comments:

  • There are several typos in the manuscript
  • Roughly half of the citations are missing in the bibliography (22-43)
  • Line 220: Students score lower in ‘Implement knowledge’ contrary to what is reported by the authors.
  • Line 299: I would suggest to re-write these brackets, e. g. (in in- and out-patient settings) as GPs are also clinicians.

Response:

We had the text re-checked by a native speaker.

Sorry for the missing citations, which is a technical error caused by Endnote Software (traveling library issue).

Thanks for the hint, we corrected this error.

Thanks again, we corrected this:

However, a review showed that physicians (both in in- and out-patient settings) encounter various obstacles when using EBP…

In conclusion, I appreciate the authors efforts to elucidate health professionals’ and students’ perception and use of EBM. However, for the latter population only very limited rationale is presented which is also mentioned by one of the participants. Why should students use EBM? For sure this is important in their clinical development later on, but at the present stage I am not able to see how they could apply that concept properly.

Further, some important socio-demographic data is missing (which health professionals? Which students?) while the relevance and definition of other variables is not clear and not followed up later on in the manuscript (‘Residence’).

The discussion remains very superficial and did not point out any consequences resulting from the present study. In terms of medical education, the authors suggest educating more on EBM but do not give any ideas how this could be achieved. Consequences for health professionals are not even discussed.

Response:

Again, we cordially thank Reviewer 1 for the many valuable comments that helped to increase the readability of the article. We hope that we addressed the issues mentioned above, especially correcting errors, deleting distracting thoughts, and increasing clarity, and we are more than happy to address further comments.

Reviewer 2 Report

The mixed usage of evidence-based medicine (EBP) and evidence-based practice (EBP) has made this article hard to follow at times.  To North American readers, EBM and EMP are not fully interchangeable terms.  EBM is generally used by the medical profession, while EBP is generally used by nursing, public health, and allied health professionals.  To further create potential confusion, the term eminence-based medicine is used without a clear definition.  Therefore, it is highly suggested to (a) consistently choose between using the terms evidence-based medicine and evidence-based practice and (b) if evidence-based medicine is chosen, avoid using the EBM acronym.  It could be confused with eminence-based medicine.  Finally, please clearly define the differences among the terminology evidence-based medicine, evidence-based practice, and eminence-based medicine.  You may even want to consider using different terminology for eminence-based medicine, in order to help limit any confusion on the part of the reader.  If the suggestion for different terminology is implemented, then citations 4, 5, and 8 may need to be dropped or replaced.

Additionally, evidence-based medicine actually began as a discipline in 1981 at Canada's McMaster University.  It did not begin during the 1990's - as stated in lines 12, 282, and 365.  This also changes the timeline used in line 31.

Your keywords (lines 27 & 28) need some work.  "eminence-based medicine" is used but not "evidence-based medicine" or "evidence-based practice".  Being that "eminence-based medicine" is not a major focus of the study, it could probably be eliminated.  Also, based on the purpose of the study, "digitalization" can also be dropped.

The research methodology that employs the use of the EBP Capability Beliefs Scale (EBPCBS) is very sound.  However, all three tables are not easyis to follow.  Maybe consider presenting them as graphs instead of tables.  Also, please consider adding a fourth graph for gender comparison.  The written-only comparison (used in Lines 221-226) is even harder to follow in the current format. 

Furthermore, the Urban vs. Rural data collected is not even analyzed within the article.  Thus including these data points in Table 1 is not necessary.

As for your references, there are two citations with incomplete information.  Number 2 only includes the subtitle of the book.  The full title is Evidence-Based Medicine: How to Practice and Teach EBM.  You also cite the first edition, where the most recent edition is the 5th (copyright 2019).  Number 4 appears to have incomplete information about the article (or is it a book?).  Finally, Number 14  seems to have a minor word processing error that should also be fixed.

If all of these suggestions are taken into account, then a rewriting of the conclusion should be strongly considered.  It is mostly opinion-based, without including strong supporting evidence from the study.